# High-latitude crochet: solar flare-induced magnetic disturbance independent from low-latitude crochet

Masatoshi Yamauchi[1], Magnar G. Johnsen[2], Carl-Fredrik Enell[3], Anders Tjulin[3], Anna Willer[4], and Dmitry A. Sormakov[5]

[1]Swedish Institute of Space Physics (IRF), Kiruna, Sweden
[2]Tromsø Geophysical Observatory (TGO), UiT the Arctic University of Norway, Tromsø, Norway
[3]EISCAT Scientific Association, Kiruna, Sweden
[4]National Space Institute, Technical University of Denmark (DTU Space), Kgs. Lyngby, Denmark
[5]Arctic and Antarctic Research Institute (AARI), St. Petersburg, Russia

**Correspondence:** M.Yamauchi (M.Yamauchi@irf.se)

**Abstract.** Solar flare-induced high latitude (peak at 70-75° geographic latitude, GGlat) ionospheric current system was studied. Right after the X9.3 flare on 6 September 2017, magnetic stations at 68-77° GGlat near local noon detected northward geomagnetic deviations ($\Delta$B) for more than 3 hours, with peak amplitudes >200 nT, without any accompanying substorm activities. From its location, this solar flare effect, or crochet, is different from previously studied ones, namely, subsolar crochet (seen at lower latitude), auroral crochet (pre-requires auroral electrojet in sunlight), or cusp crochet (seen only in the cusp). The new crochet is much more intense and longer in duration than the subsolar crochet. The long duration matches with the period of high solar X-ray flux (more than M3-class flare level). Unlike the cusp crochet, interplanetary magnetic field (IMF) $B_Y$ is not the driver, with $B_Y$ only 0-1 nT out of 3 nT total field. The equivalent ionospheric current flows eastward in a limited latitude range but extended at least 8 hours in local time (LT), forming a zonal current region equatorward of the polar cap on the geomagnetic closed region.

EISCAT radar measurements over the same region as the most intense $\Delta$B near local noon show enhancements of electron density (and hence ion-neutral ratio) at these altitudes ($\sim$100 km) where strong background ion convection (>100 m/s) in the direction of tidal-driven diurnal (Sq0) flow pre-existed. Therefore, this new zonal current can be related to this Sq0-like convection and the electron density enhancement, e.g., by descending E-region height. However, we have not found why the new crochet is found in a limited latitudinal range, and therefore the mechanism is still unclear compared to the subsolar crochet that is maintained by transient re-distribution of the electron density.

The signature is sometimes seen in the Auroral Electrojet (AE) index. A quick eye-survey for X-class flares during solar cycle 23 and 24 shows clear AU increases for about half the >X2 flares during non-substorm time, despite the unfavorable latitudinal coverage of the AE stations to detect this new crochet. Although some of these AU increases could be the auroral crochet signature, the high-latitude crochet can be a rather common feature for X flares.

**key points**

(1) We found a new type of the solar flare effect on the dayside ionospheric current at high latitudes but equatorward of the cusp during quiet periods.

(2) The effect is also seen in the AU index for nearly half of the >X2-class solar flares.

(3) A case study suggests that the new crochet is related to the Sq (tidal-driven part) current.

*Copyright statement.* TEXT

## 1   Introduction

Solar flares are known to enhance the ionospheric electron density and thus influence the electric currents in the D- and E-region. The geomagnetic disturbance ($\Delta H$, $\Delta D$, and $\Delta Z$) caused by this current system is called a "crochet" or "SFE (solar

flare effect)" (e.g., Dodson and Hedeman, 1958). Crochets are observed at dayside low-latitudes with a peak near the subsolar region (Curto et al., 1994), in the nightside high-latitude auroral region with a peak where the geomagnetic disturbance $\Delta B$ pre-exists during solar illumination (Pudovkin and Sergeev, 1977), and in the cusp (Sergeev, 1977). This paper distinguishes them by calling them "subsolar crochet", "auroral crochet", and "cusp crochet", respectively.

     The subsolar crochet is most likely caused by a re-distribution of the electron density at < 120 km altitude that is enhanced

by the flare X-ray (e.g., Curto et al., 1994; Yamasaki and Maute, 2017), resulting in a twin-vortex (one in each hemisphere) ionospheric current that is similar to the tidal-driven (daily neutral convection starting from subsolar region) part of solar quiet (Sq) ionospheric current, Sq0, which dominates low-latitude (low solar zenith angle) diurnal convection. All recent studies of the crochet (it is simply called crochet without distinction) refer to this type of disturbance. Sq also has a high-latitude part, SqP, that is driven by the solar wind-magnetosphere coupling for lowest coupling (quiet) condition, giving different current

direction from Sq0 ionospheric current (Matsushita, 1971), but SqP is found only at high geomagnetic latitudes and is not relevant to the subsolar crochet.

     The auroral crochet is most likely caused by modification of a pre-existing ionospheric current (or electrojet) by the enhanced electron density (Pudovkin, 1974). This effect increases as background plasma convection (ionospheric electric field) increases, and hence is most visible during the polar disturbances DP1 and DP2 (Akasofu, 1964; Nishida, 1968) as long as the ionosphere

is sunlit, for example, near summer solstice when X-ray flux reaches high latitudes.

     The preference of strong background plasma convection applies even to $\Delta B$ in the cusp that is strongly controlled by the interplanetary magnetic field (IMF) $B_Y$: DPY (Friis-Christensen and Wilhjelm, 1975; Levitin et al., 1982). DPY is associated with a narrow convection "throat", which is deflected eastward or westward depending on the IMF $B_Y$ polarity (Heelis et al., 1976; Yamauchi and Slapak, 2018), and hence solar flare-induced $\Delta B$ at the cusp is expected to show strong IMF $B_Y$

dependence during quiet periods. In fact, Sergeev (1977) showed one case each for both IMF polarities, showing that $\Delta B$ at > 80° latitude are in the same direction as IMF $B_Y$-dependent geomagnetic disturbances. This is the reason for calling this type of crochet "cusp crochet." Since the cusp crochet are localized to the cusp, SqP that is seen in a wider area has not been considered as important for the cusp crochet.

The amplitude and duration of the subsolar crochet is tens nT and less than 30 min (mean 16-20 min) no matter how long the high X-ray flux continues (Sergeev, 1977; Curto et al., 1994). Even after X9.3 flare on 6 September 2017, $\Delta B$ was only about 70 nT and it ended in less than 20 minutes (Curto et al., 2018), although the X-ray flux exceeded the X-flare level for nearly two hours (12:00-13:40 UT) as shown in Fig. 1a. Thus, the subsolar crochet is a transient event that corresponds to the change in the global distribution of electron density after the solar flare but not maintained by high radiation flux.

The relevant ionospheric current is expected to be limited to low and mid latitudes in the dayside, and the observed subsolar crochet amplitude actually diminishes toward terminator and high latitudes. Therefore, the subsolar crochet near the terminator has simply been assumed negligible, being driven by the weak return current of the crochet current (e.g., Annadurai et al., 2018). With the resultant day-night asymmetric nature, the subsolar crochet can be detected as a short-lived (15-20 min) spike in mid-latitude geomagnetic indices ASY-H and ASY-D (Singh et al., 2012), whereas the deviation is barely seen in high-latitude indices AU and AL except near the summer solstice (Sergeev, 1977).

Compared to the subsolar crochet, crochets at high latitudes including the auroral crochet have not been much studied for 4 decades. This is partly because crochets at high latitudes during quiet periods were considered as a simple extension of the subsolar crochets to the summer solstice, and partly because the purpose of the high-latitude crochet studies in the 1970's was to understand the ionospheric electric field during disturbed periods. Such derivation requires many assumptions (Pudovkin and Sergeev, 1977). Since the 1980's, more direct methods (satellite and radar measurements) than using geomagnetic signatures took over for the E-field studies, which led to low research activity on crochets at high latitudes even during quiet periods.

However, as shown in this paper, we found that the crochet at high latitudes is not a simple extension or sub-effect of, but is independent from the subsolar crochet with larger amplitude and longer duration. We show this from a case study of X9.3 flare on 6 September 2017, using high-latitude magnetometer data in the dayside and EISCAT radar data. We also show how these effect are seen in geomagnetic AE index using about 60 non-substorm time flares of >X2 class during past two solar cycles (cycle 23 and 24). The solar flare X-ray data observed by GOES satellites are obtained from NOAA, and the geomagnetic indices are obtained from the world data center (Kyoto and Copenhagen). The solar wind data and energetic particle data are not shown here because they are not essential to this paper and because they are described in Yamauchi et al. (2018).

## 2    High-latitude crochet for X9.3 flare on 6 September, 2017

In the overview paper of EISCAT radar observations and geomagnetic disturbances near local noon during the 6-8 September 2017 space weather event, Yamauchi et al. (2018) briefly mentioned a sudden enhancement of $\Delta B$ (> 150 nT) at high latitudes (> 68° GGlat) in response to the X9.3 flare on 6 September, 2017, but without special note nor detailed description on this high-latitude disturbance compared to subsolar crochets, auroral crochets, or cusp crochets.

### 2.1    Subsolar crochet after X9.3 flare

Fig. 1b shows detailed time profiles of the northern Scandinavian magnetograms at >65° GGlat, which are located near local noon when the X9.3 flare took place. Although the Fig. 1 period is the middle of the space weather event that started 4

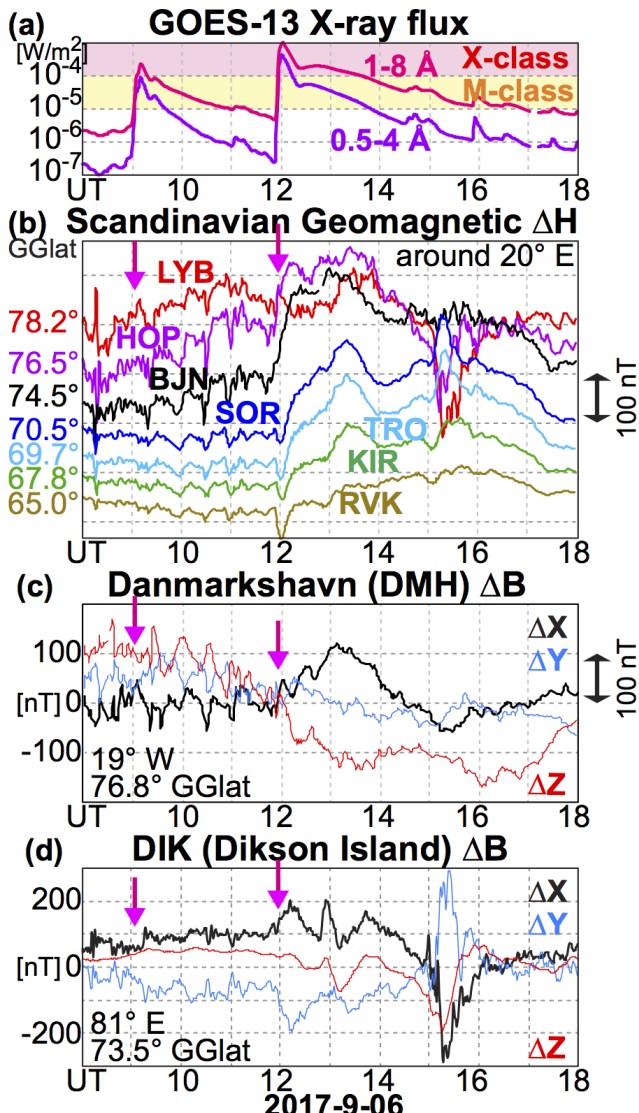

**Figure 1.** (a) Solar X-ray flux. (b) Stack-plot of geomagnetic H component for Kiruna (KIR) and northern Norway. (c) Geomagnetic data at Danmarkshavn (DMH) and (d) at Dikson Island (DIK).

September, 2017, the magnetic storm did not start until the end of 6 September. Furthermore, the solar wind during this period was stable at around 450 km/s (slightly declining in time), and IMF was weak with total field less than 3 nT ($B_X$ = -1 nT, $B_Y$ = 1 to 0 nT, $B_Z$ = -2 nT during 12:00-13:00 UT in Geocentric Solar Magnetospheric coordinate), as shown in Yamauchi et al. (2018, Fig. 1). Such stable condition caused the preceding substorm activity before the X9.3 flare diminish before the flare onset, as seen in the geomagnetic indices (Fig. 2). The IMF $B_Y$ condition indicates that the cusp crochet must be small or invisible.

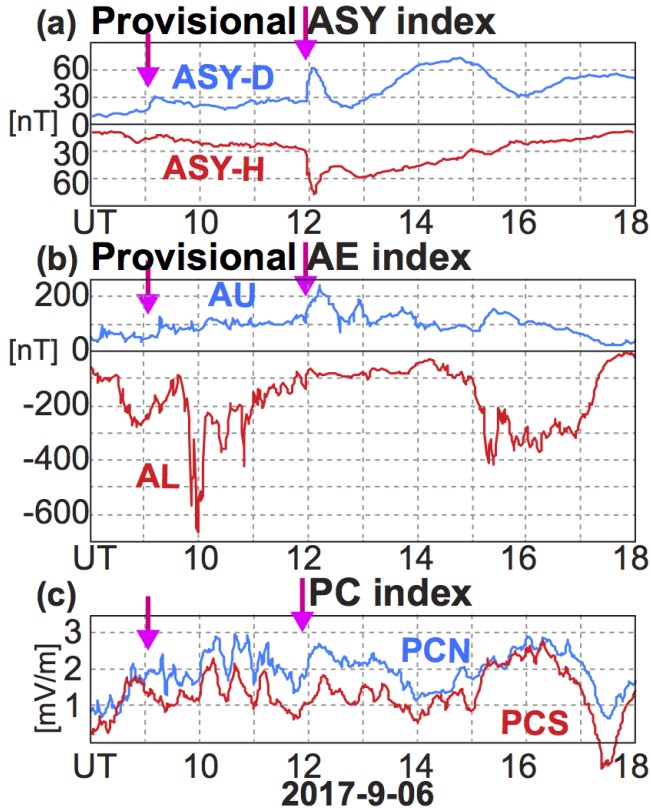

**Figure 2.** Geomagnetic ASY, AE, PC indices.

The subsolar crochet was observed as the short-lived geomagnetic deviations starting nearly simultaneously as the increase of X-ray flux at the Earth and lasted about 10-15 minutes at Sørøya (SOR, 70.5° GGlat), Tromsø (TRO, 69.7° GGlat), Kiruna (KIR, 67.8° GGlat), Rørvik (RVK, 65.0° GGlat.). All equatorward stations show the same type of disturbances (Curto et al., 2018). This is also seen in ASY-D (63 nT at 12:04 UT) and ASY-H (77 nT at 12:05 UT) as shown in Fig. 2a, with amplitude change by the flare about 60 nT. One can even recognize a crochet-like signature in ASY-D when an X2.2 flare occurred at around 09:00 UT.

## 2.2 New crochet after X9.3 flare

The new finding is the subsequent geomagnetic disturbances: a large positive ΔH deviation (northward ΔB) also started right after or even during the negative ΔH spike (south-weastward ΔB) of the subsolar crochet, with much higher amplitudes as shown in Fig. 1b. This positive ΔH continued for hours with the peak at around 13:00 UT at Bear Island (BJN) at 74.5° GGLat (> 200 nT), 13:20 UT at 70° GGlat (SOR and TRO: 180 nT) and 68° GGlat (KIR: 140 nT), and 14:50 UT at 65° GGlat (ROR: 70 nT). With larger amplitude and longer duration than the subsolar crochet, this geomagnetic signature is visible even in the

AU index, as shown in Fig. 2b, although the baseline is as large as 100 nT due to the previous substorm activity and can hide the subsolar crochet if any exists.

The development is also quick. At BJN (74.5° GGlat), $\Delta H$ reached $\Delta H > 65$ nT at already 12:05 UT, i.e., at the peak time of the subsolar crochet, and reached 130 nT at 12:20 UT. Since the long duration already indicates that the generation mechanism is different from that of the subsolar crochet (re-distribution of electron), the positive $\Delta H$ of this crochet with diminishing amplitude toward lower latitude should cancel the negative $\Delta H$ of the subsolar crochet at high-latitude. In fact, $\Delta H$ exceeded the value before the flare at 12:10 UT at 70° GGlat, 12:12 UT at 68° GGlat, and 12:17 UT at 65° GGlat.

These large $\Delta H$, however, is observed in a limited latitudinal range, diminishing toward higher latitudes with 140 nT at 76.5° GGlat (Hopen: HOP, at 12:50 UT) and not visible at 78.2° GGlat (Longyearbyn: LYB). Since the geomagnetic latitude of LYB is only 75.3° and IMF is weak with $B_Y = 0$ nT, the positive $\Delta H$ is limited to the geomagnetic closed region outside the cusp nor polar cap. The closed geometry is also indicated by the EISCAT Svalbard radar data (Yamauchi et al., 2018). The PC index (Polar Cap index), which corresponds to the polar cap activity, shows enhanced values in the same period, but not as prominent as in AU, as shown in Fig. 2.

On the other hand, positive $\Delta B$ at around 75° GGlat was observed in a wide local time range, as shown in Figs. 1c and 1d ($\Delta X$ is nearly the same as $\Delta H$ in both stations). Danmarkshavn (DMH, at 19°W) and Dikson Island (DIK, at 81°E) showed $\Delta X$ from the values before the flare about 120 nT and 100 nT at around 13:00 UT, respectively. Together with the zero IMF $B_Y$ condition, the observed large $\Delta H$ cannot be a cusp crochet. Considering its location and pre-existing activity, this crochet is neither the subsolar crochet nor auroral crochet, although some part of the observed $\Delta H$ could be affected by the auroral crochet.

For example, DIK is located near the evening terminator (it is still under sunlight near horizon) and the geomagnetic activity before 12 UT indicates some auroral activity. Therefore, the first peak at around 12:20 UT, which is larger than that of BJN or HOP and with more westward $\Delta B$, can be the auroral crochet rather than the extension of the new crochet. However, the second and third peaks are in the same direction (northward $\Delta B$) as near local noon, and multiple peaks are not expected for an auroral croche under smoothly declining X-ray flux. Therefore, the positive $\Delta X$ at DIK at around 13:00 UT and 13:40 UT can be interpreted as a part of the new high-latitude crochet rather than the auroral crochet, although we cannot dismiss the possibility of the auroral crochet.

With such large amplitude, the crochet is visible even in AU although the AE stations are not located at favorable GGlat. During 12:00-14:00 UT, AU has three positive peaks (provisional values of 240 nT at 12:12 UT, 190 nT at 12:55 UT, and 160 nT at 13:43 UT, while further baseline subtraction might be needed). The timing of these peaks corresponds to the subsolar crochet and the high-latitude one at BJN, but the provisional AU value reflects DIK data (DIK is one of the AE station) as shown in Figs. 1c and 2b. Although the amplitude is larger at BJN than DIK for the second and third peaks, BJN is located far poleward of the AE stations and does not contribute to AU.

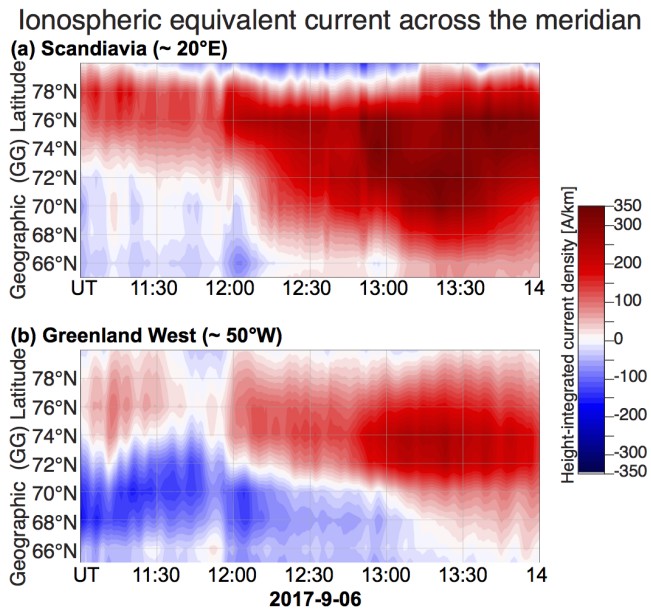

**Figure 3.** Latitudinal distribution of ionospheric equivalent current (including Sq current), Eastward component crossing (a) Scandinavian meridian (20°E) and (b) Greenland meridian (50°W), based densely distributed magnetometer network. The Spherical Elementary Current System (SECS) technique is used. Data is displayed in a latitude-time spectrogram form over 11:00-14:00 UT, 6 September 2017, i.e., around the X9.3 flare.

## 2.3 Equivalent ionospheric current

From Fennoscandian, Icelandic and Greenland magnetometer data, we also calculated the ionospheric equivalent currents (including Sq current), using the Spherical Elementary Current System (SECS) technique (Amm, 1998; Amm and Viljanen, 1999). Here, we obtained epsilon = 0.037 in a similar fashion as by Wygant et al (2012). Note that "quiet levels" that represents the internal and crustal geomagnetic field (without even Sq) were removed before applying the data to the SECS technique. They were calculated by a least-square-root approximation (rather than least squares or means), where the square root emphasizes the small variations around the quiet level, rather than larger disturbances such as substorms. To have a sufficient amount of variations with low activity for the calculation while avoiding contamination of main field secular variation, the removal of the quiet levels is performed on 10 days of data centered on the day of interest. The uncertainty is estimated to be less than 10 nT (cf. Edvardsen et al. (2013) for more details).

The results are shown in Figs. 3 and 4. Fig. 3 shows latitudinal distribution of the eastward current density crossing two meridians (Scandinavia: 20°E, and Greenland: 50°W) where actual magnetometer network are deployed. In Fig. 3, one can see a sudden appearance in the ionospheric current in a wide region at around 12:00 UT when the X-ray flux from the X9.3 flare increased at the Earth. The enhancement is westward at lower latitudes (< 70° GGlat at 20°E or 13 LT, and <72°W at 50°W or 9 LT), and eastward at higher latitudes in the both meridians. They correspond to the subsolar crochet current in the

northern hemisphere (Curto et al., 2018) and the new high-latitude crochet mentioned above, respectively. Fig. 4a and 4b show the 2D vector directions, corresponding to this timing: right before the flare (11:50 UT), and at the peak of subsolar crochet (12:04 UT). The low-latitude side composes the counter-clockwise current that agrees with the return current direction of the subsolar crochet at high latitudes (Curto et al., 2018; Annadurai et al., 2018). The high-latitude side forms another independent counter-clockwise current, with strong eastward current near BJN(cf. Fig. 1) as mentioned above. The resultant shear, which is formed poleward of BJN, corresponds to upward field-aligned current, i.e., to afternoon Region 1 field-aligned current (Iijima and Potemra, 1976; Yamauchi and Slapak, 2018).

The eastward current expanded quickly in both longitudinal direction and toward lower latitude as soon as the subsolar crochet diminished. At the same time, the current density gradually increased toward its multiple peaks. Figs. 4c and 4d show the 2D vector directions at these peaks: at the first minor peak of $\Delta H$ at around 75° GGlat (12:20 UT) after the subsolar crochet disappeared, and at the major peak of $\Delta H$ at around 70° GGlat (13:20 UT), respectively. By 12:20 UT, area of this westward current, that is most intense at around 72-74° GGlat, expanded in a wide local time from Greenland to eventually Siberia (DIK at 81°E as shown in Fig. 1d), i.e., more than 130° in longitude. The entire current lies on the geomagnetic closed region as mentioned above, and its peak latitude gradually moved equatorward. By the peak time at around 13:20 UT, all region over 5° in latitude and >100° in longitude are intensified, with much higher intensity than the subsolar crochet current.

The eastward current direction (or positive $\Delta H$) is the same as the ionospheric current in the evening auroral oval (we here mean that the region between the upward Region 1 field-aligned current and downward Region 2 field-aligned current (Iijima and Potemra, 1976; Akasofu, 1977; Yamauchi and Slapak, 2018)), and the observed eastward current continued until the next substorm onset took place at 15:00 UT (cf. Fig. 2b). However, no outstanding substorm is visible in AE (Fig. 2b) or DIK data (Fig. 1c) before this substorm with continued weak IMF condition. The eastward current patch is even started at Greenland meridian at 9 LT and continued toward the afternoon sector although IMF $B_Y = 0$ nT. Since there was no auroral current signature at BJN before this crochet, this current system is not the auroral crochet current. Rather the question is how much does this new crochet contribute to $\Delta B$ in the evening sector, e.g., compared to the auroral crochet. In this sense, we cannot judge for the moment whether the crochet detected in DIK is an evening extension of this crochet or auroral crochet or both effects mixed.

## 2.4   EISCAT data

For this X9.3 flare event on 6 September 2017, the duration of this new crochet matches with that of high X-ray flux: it was at the X-class level until 13:40 UT and M-class level until about 16:50 UT, as shown in Fig. 1. From this coincidence, Yamauchi et al. (2018) speculated about the possibility of enhancement of pre-existing Sq0 (solar quiet tidal-driven) current without showing detailed data. However, the Sq0 has long been expected to be small at high-latitude (Yamasaki and Maute, 2017). Therefore, we need direct evidence with this Sq0 scenario. For that purpose, ionospheric electron density and ion velocity, both observed by EISCAT VHF radar at Tromsø, are shown in Fig. 5.

Fig. 5b shows that the electron densities in the 100-200 km altitude range were significantly enhanced by the enhanced X-ray flux, starting at around 12 UT (doubling at 100 km altitude and seen up to 200 km altitude). Fig. 5a shows that northward ion

convection was also enhanced, and more importantly that the background Sq0 ion convection (seen as large geomagnetic ΔH > 0) starting from around 9 UT is already strongly northward (away from the sub-solar region) with large values as much as 150 m/s at 71° Mlat (at 100 km altitude).

Since the increase in the electron density means increase of the ion-neutral density ratio too, the ionospheric current is expected to flow at lower altitude, where the tidal (Sq0) ion convection is stronger. Such a change can enhance the pre-existing ionospheric Sq0 current significantly, although this scenario does not easily explain why it is found in a limited latitudinal range and wide longitude.

The time development of the electron density enhancement together with the elevated ion velocity at 100 km matches with the ΔH time profile at BJN that is located under the area observed by the EISCAT VHF radar. Note that ion velocity direction (northward) is the electric field direction at this altitude where only ions are collisional with neutrals but not electrons, and hence collision-free electrons drift westward, resulting in eastward an Hall current (that causes ΔH>0 geomagnetic disturbance).

The EISCAT data in Fig. 5d also revealed a decrease of electron density above 300 km after 12 UT. Since photochemistry predicts density increase at all altitudes, this density decrease must be caused by ionospheric dynamics, such as 3D distribution of ionospheric current. This decrease does not affect the total current because, in addition to the low collision frequency that prevents conduction current, the ion velocity at > 300 km is not changing very much compared to the ion velocity at < 200 km, and not contributing to the ionospheric current.

## 3   Preliminary survey results

We further conducted quick survey of geomagnetic AXY/AE indices (cf. Figs. 3a and 3b) during past two solar cycles. There are 73 flare events with >X2 class since 1996. For all these events, we examined web-interfaced plots of the provisional AE and ASY, by adding marks that indicate the X-ray flux level and start timing, as shown in Fig. 6 for 15 July, 2002 event. We here show raw data plots that are also found as the supplemental file.

Table 1 shows the survey result. There are about 10 events that are during substorms, and most of AU/AL variation is too large to judge whether the variation is due to the flare or not, although crochet is still outstanding in ASY for half of these cases. Among the remaining 63 cases, crochets are detected in ASY almost always, and the exceptions (5 cases) might be attributed to non-favorable distribution of ASY stations in terms of local time and GGlat at the time of flare (UT and season). Since auroral crochets and cusp crochets do not contribute to ASY, they are interpreted as subsolar crochets. In addition, crochets are detected in AU and AL for a substantial part of the cases. From the latitude of AE stations, they are either auroral crochet or this new high-latitude crochet.

For auroral crochets, the precondition requirement is severe: a substantial auroral electrojet must pre-exist in the sunlit hemisphere. This removes more than half the cases, and therefore we expect that the new high-latitude crochet can also be observed in the AE index, as seen in Fig. 3b and Fig. 6. In Fig. 6, even AL deviation started simultaneously as the crochet. Then the question is if this AL signature is related to the crochet or not. In this example, an X3.0 flare started at 19:59 UT, with X-ray flux reaching M3-class flare level at around 20:05 UT while the solar wind and IMF were stable. AE and ASY show a

**Table 1.** Survey of 73 flare events with >X2 class

| signature | yes (@onset)[*1] | unclear | no | void[*2] | total |
|-----------|------------------|---------|-----|----------|-------|
| in ASY | 57 (5) | 6 | 5 | 5 | 73 |
| in AU | 25 (5) | 7 | 30 | 11 | 73 |
| in AL | 16 (5) | 9 | 38 | 10 | 73 |

*1: Simultaneous with onset

*2: Variation ( e.g., by substorm) is too large to separate the direct flare effect

quiet condition before the flare, and all components (AU, AL, ASY-D, and ASY-H) showed sudden changes at 20:04 UT. The signature is not short-lived that is typical to the subsolar crochet.

To examine it further, Fig. 7 shows the geomagnetic data on 15 July 2002 at the same stations as Fig. 1. We also added Kullorsuaq (KUV) data from the west coast of Greenland that is 18 MLT at the time of crochet. In Fig. 7, sudden increases of $\Delta H$ at around 20:04 UT is recognized at all stations. In addition, a negative spike started 20:07 UT at BJN, 20:13 UT at SOR, and at 20:15 UT at TRO. Except for the duration, the $\Delta H>0$ enhancement at high latitude is similar to what we observed in the 6 September 2017 event (Fig. 1). The short duration is not surprising considering the short duration of high X-ray flux (> M3

class) as indicated in Fig. 6 and the high solar zenith angle (it was about 63° in the Greenland) for this UT.

Since KUV's local time is only 17 LT, which is within the zonal extend of crochet according to DIK's data in the 6 September 2017 event (Fig. 1), this $\Delta H>0$ is quite likely to be the new high-latitude crochet. Then the new high-latitude crochet extends toward the evening quite wide, which is consistent with the season near the summer solstice. Even DIK data (which is located at 01 LT past the midnight but still under sunlight) showed a minor signature. This suggests that AU signature could be caused

by this crochet rather than auroral crochet.

On the other hand, unique bipolar signature ,where the $\Delta H>0$ period is very short, is seen at BJN. This is a candidate for the auroral crochet. In addition, $\Delta B>0$ at SOR, TRO, and DIK can be read as the disruption of the substorm-related large magnetic bay. In fact, a signature of small auroral electrojet is seen at BJN, SOR, TRO, and DIK before the crochet (starting at around 19:40 UT). Although the value at BJN returned normal and the signature is not visible at Kiruna, weak aurora existed in this

narrow region before the crochet.

However, the auroral electric field before the crochet must be very weak compared to what was reported as the auroral crochet (Pudovkin and Sergeev, 1977), and at least the $\Delta H>0$ signature, that is consistently observed at many stations with quiet pre-condition, is better interpreted as the new crochet. Then, we can even wonder if the interaction between the new crochet and the auroral oval accelerated the large $\Delta B<0$ bay.

## 4 Discussion and future tasks

### 4.1 Why not found in the past?

Although magnetic stations have been extended toward higher latitudes beyond 68° GGlat since 1980's, this new crochet has never been reported, at least not to our knowledge. One possible reason is that the phenomenon is limited to a relatively small range in geographic latitude (68°-75° GGlat) while the station should be completely outside the geomagnetic cusp (< 75° Mlat). This criterion excludes many geomagnetic stations over Greenland and Canada from finding the new crochet.

### 4.2 Need solid statistics and global perspective

There are many questions to answer on this phenomenon. One obvious question is the condition to occur. When we made a survey using quick plots, we compared only with the X-ray flux (red lines in Fig. 6) but other obvious factors such as the season and geomagnetic latitude are not sorted out. We thus need to make more solid analyses. For example, we need to include amplitudes (we examined only yes or no), examine the closure of the ionospheric current system at high latitudes, analyze global magnetometer network data, and take such statistics. Such a global perspective would also tell for which conditions the effect can be detected in AU or AL.

One important note is that the difference between the GGlat and Mlat (i.e., UT dependence) must be considered. Also, we have to note that the current system might be different between different events (size of flare may affect the intensity and size, season may affect the distribution pattern and profiles of the solar zenith angles of magnetic stations). In addition, if intensification of the Sq current is important, the new crochet might be enhanced near the equinoxes (rather than summer solstice) through inter-hemispheric coupling (Yamasaki and Maute, 2017), which avoids saturation of convection-driven charges. Therefore, it might be difficult to obtain consistent results, but at least a common feature can be obtained.

### 4.3 What is the main driver of the new crochet?

As shown in Fig. 4, this current system might be related to the enhancement of Sq0-like background current through the enhancement of both the ion/electron density and ion velocity (Pederson electric field). While the density enhancement is explained by the flare radiation, the direct cause of velocity enhancement is not clear. In addition to this obvious question, we need to know the relative importance of these enhancements on the high-latitude crochet, and need to understand the relation to the nightside crochet that we found a substantial number of cases. This requires identifying the criterion how to classify the observed crochet as the auroral crochet or nightside extension of the new crochet, which we have not yet found.

To answer these questions, we need similar radar data for different events. Since the availability of radar data in favorable observation modes and geometry (such as the EISCAT data on the 6 September 2017) is limited, we need other radar data including future facility such as EISCAT 3D for a solid answer. Such a work also probably gives some hints why the electron density decreased at > 300 km in Fig. 4. Future satellite missions that cover ionospheric E-region, such as Daedalus (Sarris et al., 2020), are strongly needed.

### 4.4 Can crochet trigger a substorm or M-I coupling?

In the preliminary survey, we found many "coincidence" cases in which a large gradient of $\Delta H$ occurred at the same time as a substorm onset or sudden intensification of a substorm when the X-flare took place. Fig. 8 shows one such example when X6.2 flare took place at 19:13 UT, and substorm onset took place immediately after. This substorm is most likely associated with the southward IMF period during 18:29–18:57 UT (not shown here) that is detected at the Sun-Earth first Lagrange point L1 by Advanced Composition Explorer (ACE), but the triggering mechanism is often not associated with IMF (e.g., Yamauchi et al., 2006). Solar wind density and velocity were stable after CME arrival 7 hours before, making the crochet as a possible trigger. Larger AU than AL right after the onset is also in agreement with crochet rather the substorm because substorm onset is characterized by large negative $\Delta H$ and $|AL| \gg AU$.

Theoretically, the crochet mechanism may trigger a substorm through sudden intensification of ionospheric density and electric field through magnetosphere-ionosphere (MI) coupling (e.g., Kan et al., 1988). Since several different onset mechanisms may cause a substorm, it is quite possible that crochet may also trigger a substorms, as one of minor onset mechanisms (Yamauchi, 2019). The investigation of such a scenario is a future task.

The same question arises with respect to the Magnetosphere-Ionosphere (M-I) coupling. The latitude range of the new crochet is inside geomagnetically closed region (near local noon), but yet close to the dayside Region 1 field-aligned current. This means that the new crochet might influence the field-aligned current system. Such a study requires satellite observation at right location and right timing.

### 4.5 Modulation of Pc5?

If the long lasting high X-ray flux influences the ionospheric current, some effect might arise from the X2.2 flare at 9 UT on 6 September 2017, i.e., on the same day as the X9.3 flare (cf. Fig. 1a). One candicate is the density moderation synchronizing with the Pc5 pulsation during the recovery phase of the substorm that started at 09:37 UT with peak at 05:58 UT with AL=-666 nT (Fig. 2). In fact, $\Delta H$ showed large-amplitude oscillations with periodicity about 30 min during 09:30-11:20 UT (Fig. 1b). However, electron density at 150-200 km altitude showed irregular oscillations with a different periodicity ((~15 min) although no modulation is seen at 100 km altitude. The periodicity in the ion convection is also about 15 min for 150-200 km altitude. The only candidate that may match with 30 min periodicity is irregularity in the ion velocity at 100 km, but profile does not really match with the $\Delta H$ variation. This suggests that the Pc5 pulsation can be modulated by the density variation at 150-200 km altitude.

### 4.6 Relation to space weather

The large extra $\Delta H$ at 68-75° GGlat might, if it happens a during substorm, even be relevant to the space weather hazard, because a large sunspot may cause coincident occurrence of strong solar flare and large substorm powered by the coronal mass ejection (CME). If the crochet mechanism can interact with a substorm and reinforce each other, and if the strong solar flare

takes place within an hour after CME hits the Earth, we expect extremely strong ionospheric current and resultant ground induced currents (GIC) that are hazardous. Therefore, crochet study is potentially related to any other research field.

## 5 Conclusions

Using magnetometer data from Northern Europe, Russia and Greenland as well as EISCAT data, we found a new type of solar flare effect on the geomagnetic disturbance (SFE, or crochet) in response to the X9.3 flare on 6 September 2017 at high latitudes (65-75° GGlat). The new crochet is located at higher latitude than the subsolar crochet (cf., Figs. 4b and 4d), and is found over a wide local time range including local noon but outside the cusp, i.e., in the geomagnetic closed region. It lasted for a longer duration with higher peak amplitude than the subsolar crochet. The equivalent ionospheric current flows eastward in a limited latitude range but extended over at least 8 hours in local time (LT), forming a zonal current region at around 70-75° GGlat (equatorward of the polar cap at least in dayside). Considering its location and duration, this crochet is different from previously studied crochets (subsolar, auroral, and cusp).

Ionospheric parameters at local noon during this crochet shows strong background ion convection before the crochet as well as sharp enhancement of electron density (and hence the ion-neutral density ratio). Thus, the new crochet can be related to increase electron density at 100-150 km altitudes, where strong background (Sq0) ion convection exists. For example, change in the E-layer height can actually cause the ionospheric current at high-latitude, but such scenario does not easily explain why it is found in a limited latitudinal range, and therefore the mechanism is still unclear.

We also examined the crochet signatures in AE and ASY indices for all X flares (> X2.0) over past two solar cycles. While the subsolar crochet is well recognized in ASY indices, crochet signatures that represent the new crochet or auroral crochet are also recognized in AU for half the cases, and even in AL index sometimes. However, AE alone cannot distinguish between this new crochet and the auroral crochet in the evening sector, and further studies is needed to understand the current system related to these crochets.

*Data availability.* The X-flare list is obtained from National Oceanic and Atmospheric Administration Space Weather Prediction Center (NOAA/SWPC) at

https://www.ngdc.noaa.gov/stp/solar/solarflares.html

and the X-ray data during these flare are obtained from

https://www.ngdc.noaa.gov/stp/satellite/goes/dataaccess.html

through plot at

https://www.polarlicht-vorhersage.de/goes-archive

that is created by Andreas Møller. The solar wind and IMF data are provided by the ACE/SWEPAM and ACE/MAG team and are obtained from the ACE Science Center website:

http://www.srl.caltech.edu/ACE/ASC/level2/index.html

and the solar wind OMNI data are obtained from NASA OMNIWeb site at

http://omniweb.gsfc.nasa.gov/ow.html

AE and ASY indices (both ASCII data and web-interfaced plots) are obtained from World Data Center for Geomagnetism (WDC Kyoto) at

http://wdc.kugi.kyoto-u.ac.jp/aeasy/index.html

In these web-interfaced plots which are found in the supplementary file, Kp is also indicated. This is provided by GFZ, Adolf-Schmidt-Observatory Niemegk, Germany. Geomagnetic data are available through Tromsø Geophysical Observatory (TGO) site, SuperMAG site, and

IRF site

https://flux.phys.uit.no/geomag.html

http://supermag.jhuapl.edu/mag/?fideli

http://www2.irf.se/maggraphs/iaga/

The EISCAT common programme data is available at:

https://portal.eiscat.se/madrigal/

*Author contributions.* MY (first author) made all contribution. MJ provided Norwegian data and made Figures 3 and 4, and relevant interpretation. CE and AT calibrated EISCAT data and its interpretation. AW provided Greenland data and its interpretation. DS provided Diikson Island data and its interpretation. All contributed in constructing the argument that the phenomena is new.

*Competing interests.* no competing interests are present

*Acknowledgements.* We thank WDC kyoto, NOAA/SWPC, and SuperMAG for processed dataset. We thank Masahiko Takeda for general information on Sq. We thank the institutes who maintain the IMAGE Magnetometer Array to obtain Fig2. 3 and 4: Finnish Meteorological Institute (Finland), Institute of Geophysics Polish Academy of Sciences (Poland), GFZ German Research Centre for Geosciences (Germany), Geological Survey of Sweden (Sweden), Sodankylä Geophysical Observatory of the University of Oulu (Finland), Polar Geophysical Institute (Russia), as well as TGO (Norway) and IRF (Sweden). University of Iceland is thanked for providing data from the Leirvogur (LRV) magnetic

observatory. EISCAT is an international association supported by research organisations in China (CRIRP), Finland (SA), Japan (NIPR and ISEE), Norway (NFR), Sweden (VR), and the United Kingdom (UKRI). MY thanks IRF magnetometer team for acquisition of Kiruna data.

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

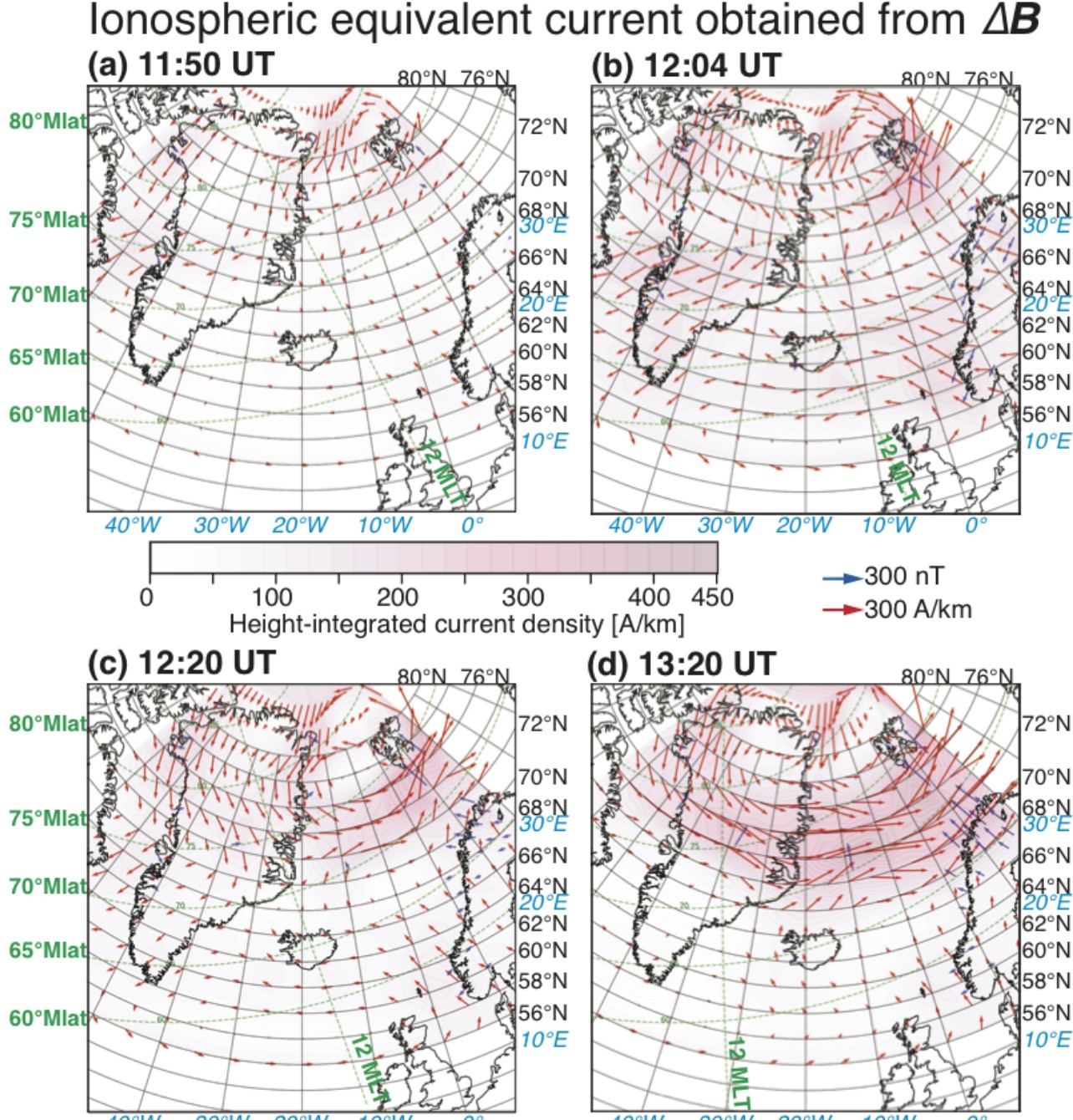

**Figure 4.** Ionospheric equivalent currents on 6 September 2017 based on Norwegian and Greenland magnetometers, using the same method as Fig. 3. (a) Before the flare (11:50 UT), (b) the subsolar crochet peak (12:04 UT), (c) around the first peak at HOP and BJN (12:20 UT), (d) the main peak at SOR, TRO, and KIR (13:20 UT).

**Figure 5.** Ionospheric line-of-sight ion velocity and electron density observed by EISCAT Tromsø VHF radar from 69° GGlat looking northward at its lowest elevation angle (30°) above the horizon. Upper panels (a) and (c) show the ion velocity and lower panels (b) and (d) show the electron density.

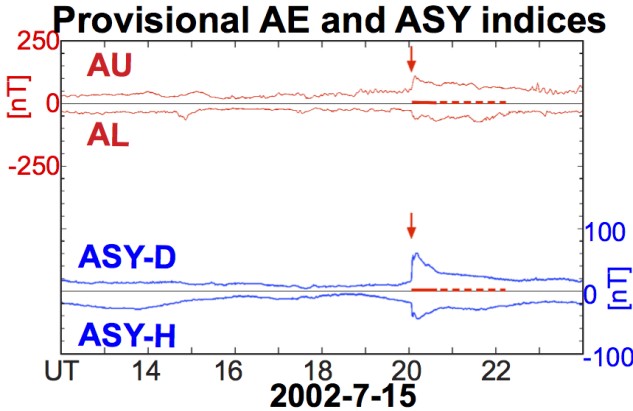

**Figure 6.** Web-interfaced plots of provisional AE and ASY indices for 15 July, 2002. The red horizontal lines correspond to the period when the X-ray flux exceeds $3 \cdot 10\text{-}5 \ \text{W/m}^2$, (>M3 class: solid lines) and $10\text{-}5 \ \text{W/m}^2$, (>M class: dashed lines). The vertical red arrow denotes the start of the crochet.

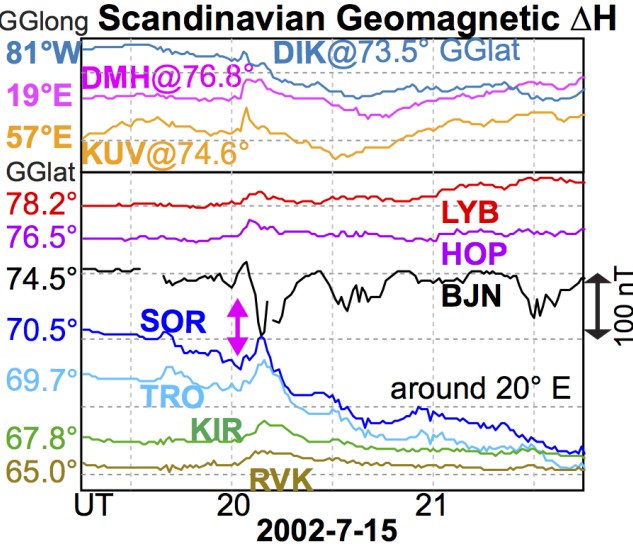

**Figure 7.** Geomagnetic ΔH in northern Scandinavia and three other stations at around 75° GGlat (Stations at Fig. 1 and Kullorsuaq: KUV) for the X3.0 flare on 15 July 2002. DIK, DMH, and KUV values are approximated by the change in local magnetic north direction. Red arrows denote the start of the X flare and subsolar crochet. At 20 UT, all stations are under sunlight at the ionosphere.

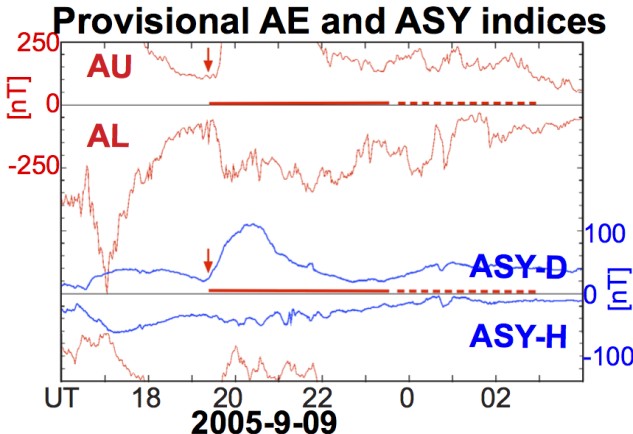

**Figure 8.** Same format as Fig. 6 for 9 September 2005. A CME arrival and southward IMF turning at the Earth are at around 12:30 UT, and 19:30 UT, respectively.