# Peer review of "High-latitude crochet: solar flare-induced magnetic disturbance independent from low-latitude crochet"

_Annales Geophysicae, 2020_

## Referee Comment (RC1) · Anonymous Referee #1 · 4 Aug 2020

High-latitude crochet: solar flare-induced magnetic disturbance independent from low-latitude By Yamauchi et al.

The paper presents a new type of the solar flare effect on the dayside ionospheric current at high latitudes equatorward of the cusp during quiet periods. Right after the X9.3 flare on 6 September 2017, magnetic stations at 68-77◦ geographic latitudes near local noon detected northward geomagnetic deviations ($\Delta$B) for more than 3 hours, with peak amplitudes >200 nT, without any accompanying substorm activities.

The paper is interesting and may be accepted for publication after addressing the minor comments below.

[Figure]

There could be many solar flares of this type. Then how this particular one produced such a large ionospheric current lasting over 3 hours and producing peak ∆B >200 nT?

Title says 'independent from low-latitudes'. But the effect is also observed in ASY indices (Figure 2a), is it consistent with the title?

Figures 1-3 are included with the text and other Figures are put at the end. It would easy if all Figures go with the text.
* * *

---

## Author Comment (AC1) · 11 Aug 2020

Thank you for the encouraging comment. We first corrected section numbering (3.1 => 4, 3.2 => 4.1, 4 => 4.2, 4.1 => 4.3, so on).

>The paper presents a new type of the solar flare effect > on the dayside ionospheric current at high latitudes > equatorward of the cusp during quiet periods. Right after > the X9.3 flare on 6 September 2017, magnetic stations > at 68-77° geographic latitudes near local noon detected > northward geomagnetic deviations ($\Delta$B) for more than> > 3 hours, with peak amplitudes >200 nT, without any > accompanying substorm activities.

[Figure]

>The paper is interesting and may be accepted for > publication after addressing the minor comments below.

The below is our answer.

>There could be many solar flares of this type. >Then how this particular one produced such a large ionospheric > current lasting over 3 hours and producing peak $\Delta B$ >200 nT?

This is the question we still have no solid answer yet. Obvious candidates are "duration of high radiation flux at high-latitudes" that is indicated from good correspondence between the duration of high radiation flux in the ionosphere, which depends on the intensity of flare itself (we examined X-ray flux but it could be different wavelength), the solar zenith angle Z (more precisely, cos(Z)), and background conductivity (as is suggested from substorm-related enhancement).

To make this question clear, we add at the top of "§4.2. Need solid statistics and global perspective (section was miss-marked as §4), something like: "There are many questions to answer on this phenomenon. One obvious question is the condition to occur. When we made a survey using quick plots, we compared only with the X-ray flux (red lines in Figure 6) but solar zenith angles, season, and geomagnetic latitude must also be examined."

To answer this, we at least need to examine all X flares and geomagnetic disturbances at high-latitude (around 70-75° geographic latitudes). So far, we have looked Norwegian geomagnetic plots (to avoid dipole tilt effect from the statistics) for all >X2 class flares since 1996, but could not conclude anything because of too few X flares when we limit season and UT (to remove solar zenith angle effect). One solution to increase the statistics to examine global network data covering different local times, but such a work is time consuming and beyond the scope of the current work (we plan to do this as the next study).

Instead, we looked at AU index for over 70 events (Table 1), and have an impression that the duration of high-latitude crochets have a relative good coincidence with the duration when the X-ray flux is >M3 level. However, this impression is quite difficult to quantify and we therefore chose mention the coincidence only for the 2017-9-6 event.

>Title says 'independent from low-latitudes'. >But the effect is also observed in ASY indices (Figure 2a), > is it consistent with the title?

Yes, it is new because the deviation observed by ASY represents the low- and mid-latitude ionospheric current system (blue color in Figure 3a and westward arrow in figure 4b) which is independent (i.e., opposite flowing direction) from the current system detected at high-latitude (red color in Figure 3a and eastward arrow in figure 4d). We will mention it in the conclusion like "is located at higher latitude than the subsolar crochet (cf., Figs. 4b and 4d)"

>Figures 1-3 are included with the text and other Figures are put > at the end. It would easy if all Figures go with the text.

Unfortunately, Annales Geophysicae's Latex package (to produce the manuscript) could not put these figures in the text, but this technical problem is definitely solved when publishing.

---

## Referee Comment (RC2) · Anonymous Referee #2 · 18 Aug 2020

Review on "High-latitude crochet: solar flare-induced magnetic disturbance independent from low-latitude" by Yamauchi et al.

A crochet is a type of geomagnetic disturbance that is typically observed at low and middle latitudes following a solar flare. This paper describes characteristics of a new type of geomagnetic crochet at high latitudes (65-75N). It is shown that the new crochet differs from ordinary ones at lower latitudes in terms of its intensity and duration. The new crochet is also shown to be different from previously reported crochets in the auroral and cusp regions. The paper contains new and exciting results that make a good addition to the understanding of the geomagnetic field. As such, I recommend this paper for publication. Below are my comments and suggestions that could further improve the quality of the paper.

1. "Sq (ion) convection" (l.12,13,167,171,172,282), "Sq" (l.236)
In my opinion, the term Sq should not be used when referring to quiet-day electric fields or currents at high latitudes. Sq electric fields and currents at middle and low latitudes are produced by the wind dynamo. At high latitudes, daily variations in electric fields and currents are not due to the wind dynamo but due to the magnetospheric convection, thus calling them Sq can be confusing. My suggestion is as follows:
l.12 Remove "Sq".
l.13 Replace "Sq" with "background".
l.167 Remove "Sq".
l.171 Replace "Sq" with "background".
l.172 Remove "Sq".
l.236 Replace "Sq" with "background".
l.282 Replace "Sq" with "background".

2. l.1 "Solar flare-induced High latitude"
"High" should be in the lower case.

3. l.36 "it is simple called crochet"
Replace "simple" with "simply".

4. l.70 "The other data are described"
What are "the other data"?

5. l.126 "Equivalent ionospheric current"
Please briefly describe how the baseline was determined. The baseline matters for equivalent currents.

6. l.162 "daily neutral convection starting from subsolar region"
This entire phrase can be replaced by "tidal winds".

7. l.164 "EISCAT VHF radar"
What is the antenna direction? The Figure 5 caption says that the radar was looking northward with 30° angle. Is this from the vertical or local magnetic field line, or something else?

8. Table 1
I do not understand this table. For example, for ASY, I see that a

crochet was detected in 52 flare events; not detected in 5 events; and unclear in 6 events. Additionally, there were 5 events where a crochet was unclear because of substorm-related disturbances. But they do not add up to the total 73 events. "+5" in the "yes" category is unexplained.

Also, it is strange to see that the number of "substorm" is different for ASY, AU, and AL. Would not it be more straighforward if the table is created only for the 62 events which are not concurrent with a substorm?

9. l.187 "There are about 10 events are during substorms"
Insert "that" between "events" and "are".

10. l.192 "they are either auroral"
Or what?

11. l.208 "This suggest that AU signature is most likely caused by this crochet rather than auroral crochet."
This is difficult to say without data from other LT. I suggest to replace "is most likely" with "could be". Also, replace "suggest" with "suggests".

12. l.232 "if intensification of the Sq current is important, the new crochet might be the equinox phenomenon"
This may be removed. Sq currents at middle and low latitudes exist not only during equinox but also during solstice.

13. l.236 "through the enhancement of both the ion/electron density and ion velocity"
The enhancement of plasma density can be understood as a result of increased ionization during the solar flare, but how do the authors explain the enhancement of ion velocity (i.e., electric field)?

14. l.242 "Such a work also probably give some hints"
Replace "give" with "gives".

15. l.249 "and traditional explanation of the trigger is IMF changes"
I do not understand what was meant by this. Remove or rephrase.

16. l.261 "on that day"
Please clarify which day.

17. l.264 "mediation"
"modulation"?

18. l.268 "4.4 Relation to space weather"
This subsection, consisting of two sentences, can be removed. It does not add any new information or insight.

Finally, please check the numbering of sections and subsections, which is currently as follows:
1. Introduction

2. High-latitude crochet for X9.3 flare on 6 September, 2017
 2.1 Subsolar crochet after X9.3 flare
 2.2 New crochet after X9.3 flare
 2.3 Equivalent ionospheric current
 2.4 EISCAT data
3. Preliminary survey results
 3.1 Discussion and future tasks
 3.2 Why not found in the past?
4. Need solid statistics and global perspective
 4.1 What is the main driver of the new crochet?
 4.2 Can crochet trigger a substorm or M-I coupling?
 4.3 Modulation of Pc5?
 4.4 Relation to space weather
5. Conclusions

Subsection 3.1 has no content. Perhaps, Subsection 3.1 was meant to
be Section 4, and
3.2 -> 4.1
4. -> 4.2
4.1 -> 4.3 and so on?

---

## Author Response (AR1)

Dear Dalia

We have improved the manuscript along the way that we have replied to both reviewer's comments on the open discussion (copy below, and red-marked in the manuscript). We also generally improved English expression. We also uploaded the file of all web-interfaced plots of AE and ASY that was promised in line 205 (blue marked).

In addition, we have mentioned the context of this period in the space weather events 4-12 September, 2017 in §2.1. In short, no effect is seen.

Sincerely
Yama

======================
Reply to Referee #1

Thank you for your encouraging comment.

**reviewer**
The paper presents a new type of the solar flare effect on the dayside ionospheric current at high latitudes equatorward of the cusp during quiet periods. Right after the X9.3 flare on 6 September 2017, magnetic stations at 68-77° geographic latitudes near local noon detected northward geomagnetic deviations ($\Delta B$) for more than 3 hours, with peak amplitudes >200 nT, without any accompanying substorm activities. The paper is interesting and may be accepted for publication after addressing the minor comments below.

**comment 1# There could be many solar flares of this type. Then how this particular one produced such a large ionospheric current lasting over 3 hours and producing peak $\Delta B$ >200 nT?**

**Answer# This is the question we still could not have solid answer.**

To answer this, we have to examine all X flares and geomagnetic stations disturbances at high-latitude (around 70-75° geographic latitudes). We tried this with Norwegian data (to remove dipole tilt effect from statistics) for all >X2 class flare, but could not answer because there are not many X flares if we limit season and UT (to remove solar zenith angle effect). To have sufficient examples, we need to examine global network data covering different local times, which is too big work to finish soon (we plan to perform as the next study).

Instead we took statistics with AU index, and have a feeling that duration of high-latitude Crochet is relatively good coincidence with the duration when the X-ray flux is >M3 level, but this feeling is quit difficult to quantify and did not mention in the manuscript.

**comment 2# Title says 'independent from low-latitudes'. But the effect is also observed in ASY indices (Figure 2a), is it consistent with the title?**

**Answer# Yes it is new because deviation observed by ASY represents low- and mid-latitude ionospheric current system (blue color in Figure 3a and westward arrow in figure 4b) that is independent (i.e., opposite flowing direction) from high-latitude current system that is detected at high-latitude (red color in Figure 3a and eastward arrow in figure 4d).**

**comment 3# Figures 1-3 are included with the text and other Figures are put at the end. It would easy if all Figures go with the text.**

**Answer# Unfortunately, Annales Geophysicae's Latex package (to produce manuscript) could not put these figures in the text, but this technical problem is definitely solved when publishing.**

=====================
Reply to Referee #2

Thank you for your kind and constructive comments.

**reviewer**
A crochet is a type of geomagnetic disturbance that is typically observed at low and middle latitudes following a solar flare. This paper describes characteristics of a new type of geomagnetic crochet at high latitudes (65-75N). It is shown that the new crochet differs from ordinary ones at lower latitudes in terms of its intensity and duration. The new crochet is also shown to be different from previously reported crochets in the auroral and cusp regions. The paper contains new and exciting results that make a good addition to the understanding of the geomagnetic field. As such, I recommend this paper for publication. Below are my comments and suggestions that could further improve the quality of the paper.

Followings are answers to your specific questions/comment (#1, 4, 5, 7, 8, 10, 12, 13, 15, 16, and 18), and these explanation are included in the revised manuscript.

**comment 1.# "Sq (ion) convection" (l.12,13,167,171,172,282), "Sq" (l.236)**
In my opinion, the term Sq should not be used when referring to quiet-day electric fields or currents at high latitudes. Sq electric fields and currents at middle and low latitudes are produced by the wind dynamo. At high latitudes, daily variations in electric fields and currents are not due to the wind dynamo but due to the magnetospheric convection, thus calling them Sq can be confusing. My suggestion is as follows:
l.12 Remove "Sq".
l.13 Replace "Sq" with "background".

l.167 Remove "Sq".
l.171 Replace "Sq" with "background".
l.172 Remove "Sq".
l.236 Replace "Sq" with "background".
l.282 Replace "Sq" with "background".

**Answer**
We realized that we did not clearly differentiate Sq0 (tidal-driven only) and SqP (mixed with solar wind driven). When we subtract background level, it is "Sq0 + SqP" ("background" is a better name), but when we discuss "enhancement of Sq, it is Sq0 ("Sq" is the better name). We change wording respectively, with additional reference by Matsushita (1958).

Matsushita, S.: Interactions Between the Ionosphere and the Magnetosphere for Sq and L Variations, Radio Sci., 6( 2), 279? 294, https://doi.org/10.1029/RS006i002p00279, 1971.

**comment 4.# l.70 "The other data are described"**
What are "the other data"?

**Answer**
It is non-essential data for this paper (ACE data, GOES energetic particle data) but readers may want to see. We explain explicitly.

**comment 5.# l.126 "Equivalent ionospheric current"**
Please briefly describe how the baseline was determined.
The baseline matters for equivalent currents.

**Answer**
We believe that the reviewer is referring "baseline" to "quiet level" (a value without Sq), i.e., internal field. (We normally use "baseline" terminology to calibration of flux gate magnetometer data, but we think the reviewer does not mean that.) Yes, we remove this "quiet level" prior to the application of the SECS technique.

The "quiet levels" that represents the internal and crustal geomagnetic field were removed before applying the data to the SECS technique. These were calculated by a least-square-root approximation (rather than least squares or means), where the square root emphasizes the small variations around the quiet level, rather than larger disturbances such as substorms. To have a sufficient amount of variations with low activity for the calculation while avoiding contamination of main field secular variation, the removal of the quiet levels is performed on 10 days of data centered on the day of interest. The uncertainty is estimated to be less than 10 nT (cf. Edvardsen et al. (2013) for more details).

Edvardsen, I., Hansen, T. L., Gjertsen, M., & Wilson, H.:
Improving the accuracy of directional wellbore surveying in the Norwegian Sea,
Soc. Petrol. Eng. Drill. Complet., 28,
https://doi.org/10.2118/159679-PA, 2013.

**comment 7.# l.164 "EISCAT VHF radar"**
What is the antenna direction? The Figure 5 caption says that
the radar was looking northward with 30° angle. Is this from
the vertical or local magnetic field line, or something else?

**Answer**
It is 30 degree "elevation" which is the lowest allowed elevation.

**comment 8.# Table 1**
I do not understand this table. For example, for ASY, I see that a
crochet was detected in 52 flare events; not detected in 5 events;
and unclear in 6 events. Additionally, there were 5 events where a
crochet was unclear because of substorm-related disturbances. But
they do not add up to the total 73 events. "+5" in the "yes"
category is unexplained.
Also, it is strange to see that the number of "substorm" is
different for ASY, AU, and AL. Would not it be more straightforward
if the table is created only for the 62 events which are not
concurrent with a substorm?

**Answer**
We are sorry for insufficient explanation.  Five case at @Onset means simultaneous
onsets  of substorm and crochet.  We change the way to show as 57 (5) instead of
52 (+5).
Substorm column means that substorm activity is too disturbed to isolate the crochet
effect.  Since substorm disturbance is stronger in AU/AE, it is more difficult to see it.
If only ASY, substorm activity is mild enough to isolate more crochet.  We add these
explanations.

**comment 10.# l.192 "they are either auroral"**
Or what?

**Answer**
"auroral crochet or this new high-latitude crochet."
(=> When we copy-and-pasted from word to Latex, copy missed last line).

**comment 12.# l.232 "if intensification of the Sq current is important, the**
new crochet might be the equinox phenomenon"
This may be removed. Sq currents at middle and low latitudes exist
not only during equinox but also during solstice.

**Answer**
What we meant here is inter-hemispheric coupling of Sq current along the
geomagnetic field.  This is important for high-latitude.  We add inter-hemispheric
aspect in the explanation.

**comment 13.# l.236 "through the enhancement of both the ion/electron density and ion velocity"**
The enhancement of plasma density can be understood as a result of increased ionization during the solar flare, but how do the authors explain the enhancement of ion velocity (i.e., electric field)?

**Answer**
Yes it is obvious question.  It is so obvious the we forgot to write this question.

**comment 15.# l.249 "and traditional explanation of the trigger is IMF changes"**
I do not understand what was meant by this. Remove or rephrase.

**Answer**
We add explanation.  Although this substorm is most likely associated with the southward IMF during 18:29-18:57 UT at the Sun-Earth first Lagrange point L1, timing of the substorm onset and large AU compared to large AL suggests that crochet could triggered the onset.

**comment 16.# l.261 "on that day"**
Please clarify which day.

**Answer**
6 September, 2017 (same day as X9.3 flare)

**comment 18.# l.268 "4.4 Relation to space weather"**
This subsection, consisting of two sentences, can be removed.
It does not add any new information or insight.

**Answer**
A large sunspot may cause coincident occurrence of strong solar flare and large substorm triggered by the coronal mass ejection (CME). If the crochet mechanism can interact with a substorm and reinforce each other, and if the strong solar flare takes place within an hour after CME hits the Earth, we expect extremely strong ionospheric current and resultant ground induced currents (GIC) that are hazardous. I add this explanation.

All other comments (typo/expression/error) are amended as suggested

**comment 2.# l.1 "Solar flare-induced High latitude"**
"High" should be in the lower case.

**comment 3.# l.36 "it is simple called crochet"**
Replace "simple" with "simply".

**comment 6.# l.162 "daily neutral convection starting from subsolar region"**
This entire phrase can be replaced by "tidal winds".

**comment 9.# l.187 "There are about 10 events are during substorms"**
Insert "that" between "events" and "are".

**comment 11.# l.208 "This suggest that AU signature is most likely caused by this crochet rather than auroral crochet."**
This is difficult to say without data from other LT. I suggest to replace "is most likely" with "could be". Also, replace "suggest" with "suggests".

**comment 14.# l.242 "Such a work also probably give some hints"**
Replace "give" with "gives".

**comment 17.# l.264 "mediation"**
"modulation"?

**comment 19.# Finally, please check the numbering of sections and subsections, which is currently as follows:**
...
...
Subsection 3.1 has no content. Perhaps, Subsection 3.1 was meant to be Section 4, and
3.2 -> 4.1
4. -> 4.2
4.1 -> 4.3 and so on?